# Apical size reduction by macropinocytosis alleviates tissue crowding

Enzo Bresteau[1], Eve E. Suva[1], Christopher Revell [2], Osama A. Hassan[1], Aline Grata[1], Jennifer Sheridan [1], Jennifer Mitchell[1], Constadina Arvanitis[1,3], Farida Korobova[1,3], Sarah Woolner [4], Oliver E. Jensen [2] & Brian Mitchell [1,5] ✉

Tissue crowding represents a critical challenge to epithelial tissues, which often respond via the irreversible process of live cell extrusion. We report that apical size reduction via macropinocytosis serves as a malleable and less destructive form of tissue remodeling that can alleviate the need for cell loss. We find that macropinocytosis is triggered by tissue crowding via mechanosensory signaling, leading to substantial internalization of apical membrane. This drives a reduction in apical surface which alleviates crowding. We report that this mechanism regulates the long-term organization of the developing epithelium and controls the timing of proliferation-induced cell extrusion. Additionally, we observe a wave of macropinocytosis in response to acute external compression. In both scenarios, inhibiting macropinocytosis induces a dramatic increase in cell extrusion suggesting cooperation between cell extrusion and macropinocytosis in response to both developmental and external compression. Our findings implicate macropinocytosis as an important regulator of dynamic epithelial remodeling.

Epithelial tissue is a conserved feature of all metazoan animals, and its barrier function is essential for multicellular life. It can serve as a mechanical barrier between organs that is capable of withstanding a variety of stresses, including physical compression from the environment as well as the significant tissue remodeling that occurs during morphogenesis. Epithelial tissue crowding, the accumulation of epithelial cells within a confined space, typically arises from cell proliferation but can be exacerbated by externally applied compression. Defects in how tissues respond to crowding are associated with aberrant cell division and tumor growth[1–4]. To alleviate tissue crowding, an epithelium must remove some of its apical surface. An important and well-studied response is cell removal via live cell extrusion[1,3,4]. Here we propose that epithelial crowding can be addressed without irreversible cell loss, by multiple cells reducing their apical size. We propose that membrane internalization via macropinocytosis drives apical size reduction to alleviate tissue crowding.

Macropinocytosis is a non-specific endocytic pathway that relies on the formation of large circular actin ruffles that protrude from the cell surface, leading to the engulfment of extracellular material into large vesicles[5,6]. Macropinocytosis has typically been associated with nutrient acquisition, receptor recycling, and immune surveillance[6]; however, recent studies have also implicated macropinocytosis in membrane repair and recycling[7,8]. More broadly, various forms of endocytosis have been shown to influence distinct aspects of morphogenesis[9–12]. However, the role of endocytic processes in regulating tissue mechanics remains largely unknown.

In this study, we utilize the surface epithelium of developing *Xenopus* embryos to propose a previously unappreciated role for membrane internalization via macropinocytosis as a mechanism for apical size reduction in response to tissue crowding. First, we report that macropinocytotic events increasingly occur during the growth of the epithelium to alleviate local crowding during development.

[1]Department of Cell and Developmental Biology, Feinberg School of Medicine, Northwestern University, Chicago, IL, USA. [2]Department of Mathematics, University of Manchester, Manchester, UK. [3]Center for Advanced Microscopy, Northwestern University, Chicago, IL, USA. [4]Manchester Cell-Matrix Centre, School of Biological Sciences, Faculty of Biology, Medicine and Health, University of Manchester, Manchester, UK. [5]Lurie Cancer Center, Northwestern University, Chicago, IL, USA. ✉e-mail: brian-mitchell@northwestern.edu

Additionally, we propose that macropinocytosis can serve as an immediate response to environmental stress. We report that waves of macropinocytotic events occur to remodel the epithelium in response to externally induced compression of the epithelium. Together, our results suggest that macropinocytosis represents a less destructive strategy for alleviating tissue crowding, reducing the need for irreversible cell extrusion.

## Results

### Constitutive macropinocytosis in Xenopus embryo epidermis

Actin ruffling has been reported to occur in a wide range of tissues under diverse physiological conditions. Actin staining as well as imaging of dynamic actin using Lifeact in late Stage (ST34+) *Xenopus* embryos revealed numerous events of circular actin ruffling on the epithelial surface (Fig. 1a, b and Supplementary Movies 1 and 2). These events initiate rapidly and terminate in 9.3 (±2.3) min (Fig. 1d). The actin ruffles are quite large, with a maximal area of 94 (±36) μm² (Fig. 1e), which represents 22.8% (±6.6%) of the total apical surface of the host cell (Fig. 1f). While these ruffles can occur anywhere on the apical surface of the cell, we find that they preferentially occur close to cell-cell junctions (Fig. 1g). Imaging embryos using scanning electron microscopy (SEM) revealed multiple phases of these events including circular ruffles protruding apically out of the cell as well as circular protrusions that appear to be closing in on themselves (Fig. 1c). These structures are similar to the actin ruffles observed during macropinocytotic events that have been described in cell culture as well as other model organisms[5,13,14]. To confirm that our observed actin ruffles lead to macropinocytosis we bathed embryos in media containing fluorescent dextran and found that the closing of circular actin ruffles indeed leads to the internalization of dextran-filled vesicles inside the cell (Fig. 1h and Supplementary Movie 3). Importantly, this occurs via the internalization of a substantial portion of the apical membrane to form a large vesicle inside the cell (Fig. 1i).

Myosins are often the driving force in generating dynamic actin-based structures[15]. Imaging with a fluorescently tagged Myosin II intrabody (SF9)[16] revealed that active Myosin II localizes outside of the circular actin ruffles (Fig. 1j, k and Supplementary Movie 4), suggesting a role for actomyosin forces in the dynamic ruffle protrusion. Accordingly, treatment with the Myosin inhibitor, Blebbistatin, resulted in a complete loss of these macropinocytotic events (Fig. 1l).

### Macropinocytosis occurs in crowded, low-tension epithelia

We observe that macropinocytosis progressively increases during development. While we rarely observe macropinocytotic events in ST34 embryos, their occurrence is dramatically increased by ST42 (Fig. 2a, b). As development progresses, proliferation within the confined epithelium leads to an increase in cell density, which correlates with higher levels of macropinocytosis (Fig. 2a, b). To test the importance of cell density on macropinocytosis, we inhibited cell proliferation with the Cyclin Dependent Kinase (CDK) inhibitor Roscovitine[17]. As expected, proliferation inhibition led to lower cell density and tissue crowding in comparison with untreated embryos (Fig. 2c). Similarly, the level of macropinocytosis was significantly lower in Roscovitine-treated embryos, suggesting that these events are induced in crowded tissues (Fig. 2c). Tissue crowding has been associated with low levels of junctional tension in the epithelium[3,18-21], suggesting that tension levels could modulate macropinocytosis. Additionally, membrane tension is a known regulator of endocytosis in general[22] and has recently been linked to macropinocytosis[13,23]. To address the importance of tensile forces on macropinocytosis, we utilized ectodermal tissue excised from ST10 embryos. This tissue can be cultured into a 3D epithelial organoid, or it can be plated onto fibronectin coated glass where it will adhere to the substrate and spread out into a 2D explant (Fig. 2d). Because cells in the 2D explant spread onto an underlying substrate they are more stretched than cells

in the 3D organoids, which we confirmed by the direct measurement of higher cell density in organoids versus explanted tissues (Fig. 2e, f and Supplementary Movies 5 and 6). Because of this stretching, cells in explanted tissues are considered to be under higher tensile stress than the cells in 3D organoids[24-27]. Quantification of macropinocytosis level revealed that organoids have a significantly higher level of macropinocytosis as compared to the explants (Fig. 2e, f and Supplementary Movies 5 and 6) consistent with a role for tensile force and tissue crowding in driving macropinocytosis.

To further investigate how tensile forces modulate macropinocytosis, we chemically manipulated membrane tension and assessed the consequence on macropinocytosis. First, we treated embryos with Mβ cyclodextrin, a small molecule known to remove cholesterol and lipid rafts from membranes that has been used to increase membrane tension[28], and we observed an -twofold decrease in macropinocytosis (Fig. 2h). In contrast, we treated embryos with Sodium Deoxycholate (NaDeoxy), a detergent that at low doses can insert into the membrane and which has been used to lower membrane tension[29]. In embryos treated with NaDeoxy we observed a significant -twofold increase in macropinocytotic events consistent with our interpretation that low membrane tension promotes macropinocytosis (Fig. 2h).

Changes in membrane tension can be sensed via mechanosensory channels of the TRP and Piezo families. To confirm that mechanosensation is an important regulator of macropinocytosis in our system, we quantified macropinocytotic events in tissues where we manipulated mechanosensation both positively and negatively. Inhibition of mechanosensory channels (Trp and Piezo) via GsMTx4[30,31] led to an -threefold increase in macropinocytosis (Fig. 2g, h). In contrast, activating Piezo1 with Yoda1[32] led to a -fourfold decrease in macropinocytotic events (Fig. 2h). Dextran internalization was observed under all drug treatment conditions confirming proper macropinocytosis (Supplementary Movie 7). The effect of membrane tension and Piezo1 manipulation on macropinocytosis are consistent with findings in both human cells and the marine organism Hydra which have shown that macropinocytosis is negatively regulated by membrane tension[13,23,33]. These results further suggest that tissue crowding induced macropinocytosis occurs via a decrease in membrane tension.

### Macropinocytosis leads to apical cell size reduction

Various forms of endocytosis have been shown to participate in membrane remodeling during morphogenesis[10-12]. We therefore wanted to assess the consequence of macropinocytosis on the organization of the apical surface during epithelial development. By tracking the apical size of individual cells during macropinocytosis, we measured a loss of approximately 10% of apical surface area (Fig. 3a, b and Supplementary Fig. 1a). While there is some variation to the change in apical size, we observe a correlation between the size of the macropinocytotic event and the reduction in apical size further supporting the idea that these events cause the reduction ($r^2 = 0.398$, Fig. 3c). Additionally, these events preferentially occur at cell-cell junctions, and we also observed a 10% decrease in the length of junctions associated with a macropinocytotic event (Fig. 3d). However, we did not observe the internalization of junction proteins as previously reported[11] (Supplementary Fig. 1b–e). Therefore, we conclude that apical size reduction is primarily driven by internalization of a large portion of the apical membrane rather than internalization of the junctions per se.

Our observations of macropinocytosis-induced apical size reduction also revealed a local remodeling of the epithelium in response to the apical size shrinkage, which should affect local tension distribution. To investigate this, we utilized the CellFit analysis that infers relative junctional tension based on tissue geometry[34,35]. We performed CellFit on frames before and after a macropinocytotic event (Supplementary Fig. 1f) and compared the changes to inferred tension in the junctions associated with a macropinocytotic event (N1), the junctions contiguous with the N1

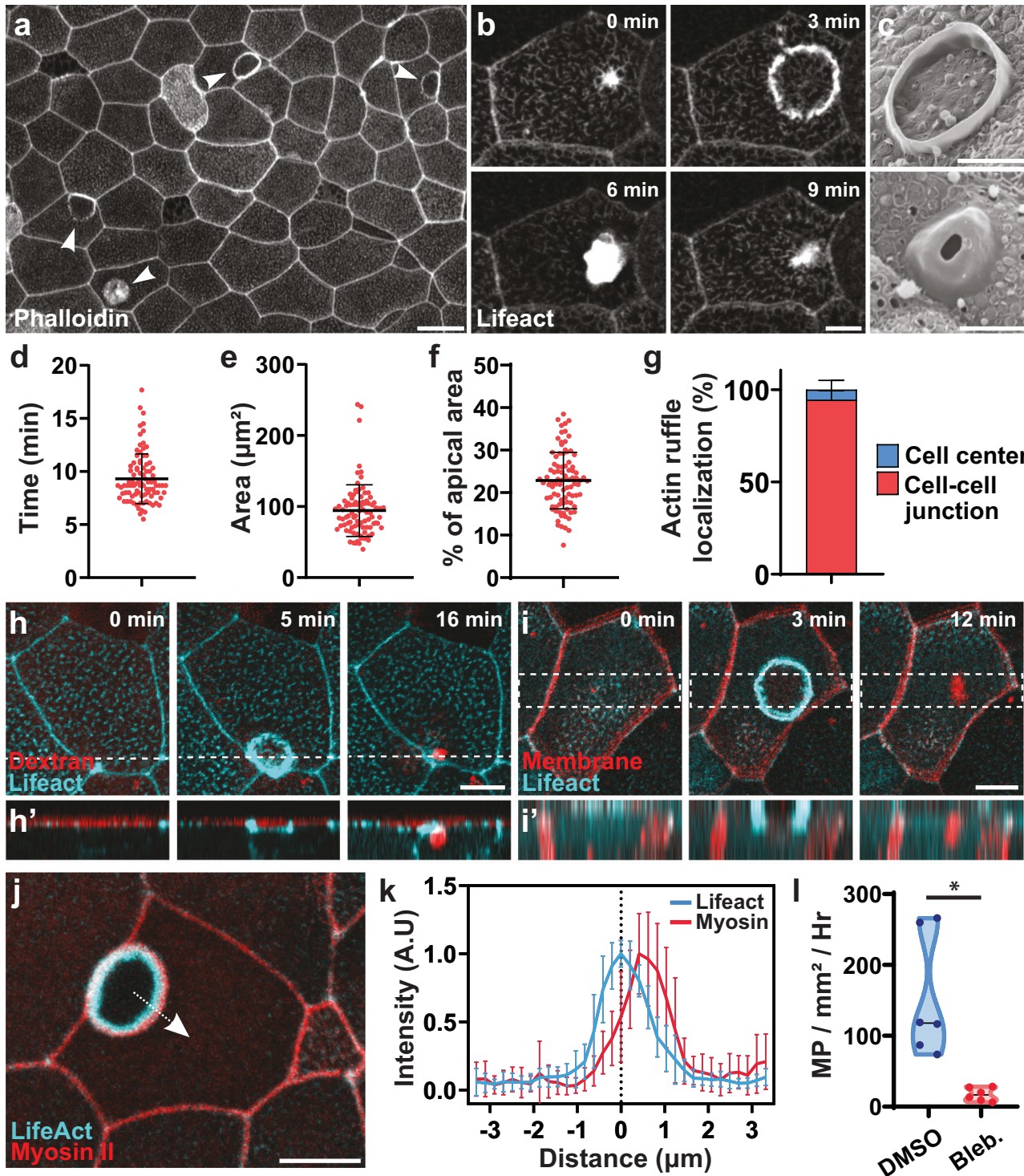

**Fig. 1 | Constitutive macropinocytosis occurs in the apical surface of the Xenopus embryonic epidermis. a** Actin ruffles (arrowheads) in the epidermis of *Xenopus* embryo stained with phalloidin. Scale bar 20 µm. **b** Timelapse of a single ruffle event in the epidermis of *Xenopus* embryo expressing Lifeact-fluorescent protein (FP). Scale bar 10 µm. **c** SEM images of actin ruffles as circular (top) or dome-shaped (bottom). Scale bar 5 µm. **d–f** Quantification of actin ruffle duration (**d**), peak area (**e**), and percentage of apical area (**f**) (*n* = 3 exp. and 90 ruffles; center line = mean; error bars = SD). **g** Quantification of actin ruffle localization within the cell (*n* = 9 exp. and 361 ruffles; bar = mean; error bars = SD). **h** Time lapse image of the Lifeact-FP (cyan) showing macropinocytotic internalization of fluorescent dextran (red) from the media, h' is a z projection of the dotted line. Scale bar 10 µm. **i** Time lapse image of the Lifeact-FP (cyan) showing internalization of apical membrane (red) during macropinocytosis, i' is a z projection of the dotted area. Scale bar 10 µm. **j** Circular actin ruffle (cyan, Lifeact-FP) surrounded by an outer ring of activated myosin II (red; SF9 intrabody). The dotted arrow shows an example of linescan averaged in (**k**). Scale bar 10 µm. **k** Localization of Lifeact and activated myosin II at the peak of an actin ruffles (*n* = 2 exp. and 15 ruffles; point = mean; error bars = SD). **l** Quantification of macropinocytosis (MP) events in embryos treated with DMSO or Blebbistatin (*n* = 3 exp. and 6 embryos; two-sided paired *t*-test, *p*-value: 0.0116). Source data are provided as a Source Data file.

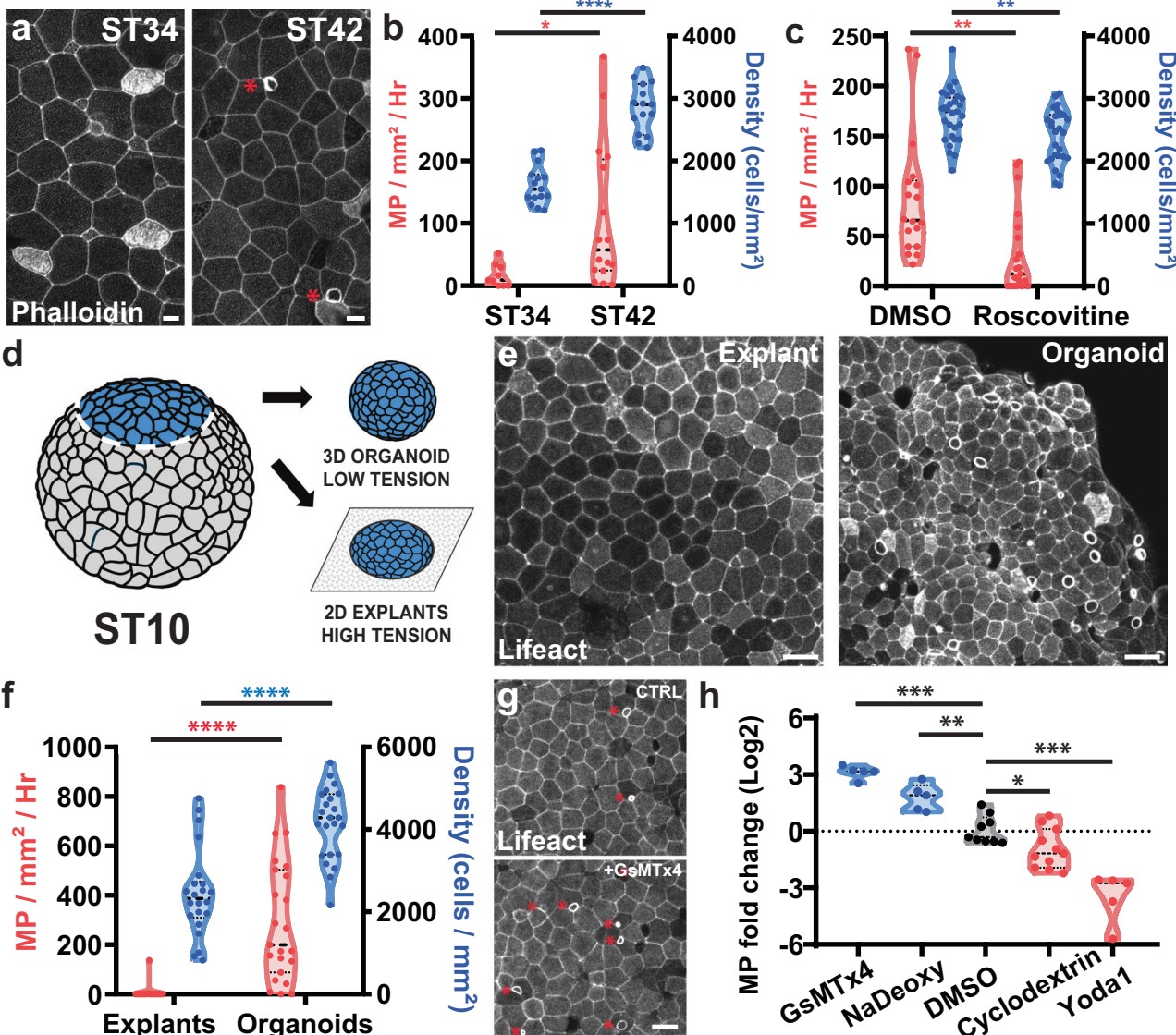

**Fig. 2 | Macropinocytosis is associated with tissue crowding and low levels of tension. a** Image of *Xenopus* stained with phalloidin (white) at ST34 (left) and ST42 (right, asterisk highlight actin ruffles). Scale bar 10 μm. **b** Quantification of cell density (blue; right axis) and macropinocytotic events (MP, red; left axis) at ST34 and ST42 (*n* = 3 exp. and 15 embryos per stage for cell density, *p*-value = 3.43e-10; 4 exp. and 6 embryos for macropinocytotic events at ST34 and 5 exp. and 16 embryos at ST42, *p*-value = 0.0282). **c** Quantification of cell density (blue; right axis) and macro-pinocytotic events (red; left axis) in embryo treated or not with Roscovitine for 24 h (*n* = 4 exp. and 18 control embryos and 21 Roscovitine-treated embryos for MP events, *p*-value = 0.0019; 3 exp. and 29 control embryos and 28 Roscovitine-treated embryos for goblet cell density, *p*-value = 0.0045). **d** Representation of the generation of 3D organoids or 2D explant plated onto fibronectin coated coverslip. Adapted from[61,62].

**e** Representative images of the surface of 2D explant (left) or 3D organoids (right) expressing Lifeact-FP. Scale bar 30 μm. **f** Quantification of cell density (blue; right axis) and macropinocytosis (red; left axis) in 2D explants or 3D organoids (*n* = 6 experiments with 22 explants and 23 organoids; MP *p*-value = 2.00e-06 and density *p*-value = 5.97e-07). **g** Image of actin ruffles (asterisk) in the epithelium of an embryo expressing Lifeact-FP before (up) and after (down) treatment with GsMTx4. Scale bar 30 μm. **h** Quantification of the fold difference in macropinocytosis level between 1 h before and 1 h after treatment with DMSO (mock treatment, *n* = 9 embryo), GsMTx4 (*n* = 5 embryo; *p*-value = 2.03e-06), NaDeoxy (*n* = 5 embryo; *p*-value = 0.0012), Mβ Cyclodextrin (*n* = 11 embryo; *p*-value = 0.0260), or Yoda1 (*n* = 5 embryo; *p*-value 3.00e-05). All tests are two-sided unpaired *t*-tests. Source data are provided as a Source Data file.

junctions (N2) and so on (Fig. 3e). This analysis revealed that relative junction tension significantly increases after a macropinocytotic event for the more proximal junctions (N1-N2), but that the effect dissipates in the more distal junctions (N3-N4) (Fig. 3f). This local increase in junction tension suggests that apical size reduction by macropinocytosis induces a local stretching of the epithelium, similar but weaker than that observed during cell extrusion[36]. From these results we propose that macro-pinocytosis can serve as a regulatory mechanism in response to crowding. Similar to cell removal by crowding-induced cell extrusion, apical- size reduction by macropinocytosis would facilitate the alleviation of tissue crowding (Fig. 3g).

## Macropinocytosis regulates crowding in epithelial development

Live cell extrusion in response to tissue crowding has been observed in numerous tissues and organisms[1,3,4]. We recently reported that in *Xenopus* embryos, a subset of multiciliated cells (MCCs) are targeted for apical cell extrusion via mechanosensation. Importantly, this extrusion increases as the embryo develops suggesting that tissue crowding is a driving factor[37]. Consistent with this, when we prevent crowding by inhibiting proliferation with Roscovitine, we observe a dramatic maintenance of MCCs compared to controls (Fig. 4a, b). Interestingly, although macropinocytotic events occur throughout the epithelium, we never observe them in MCCs, but instead have found a

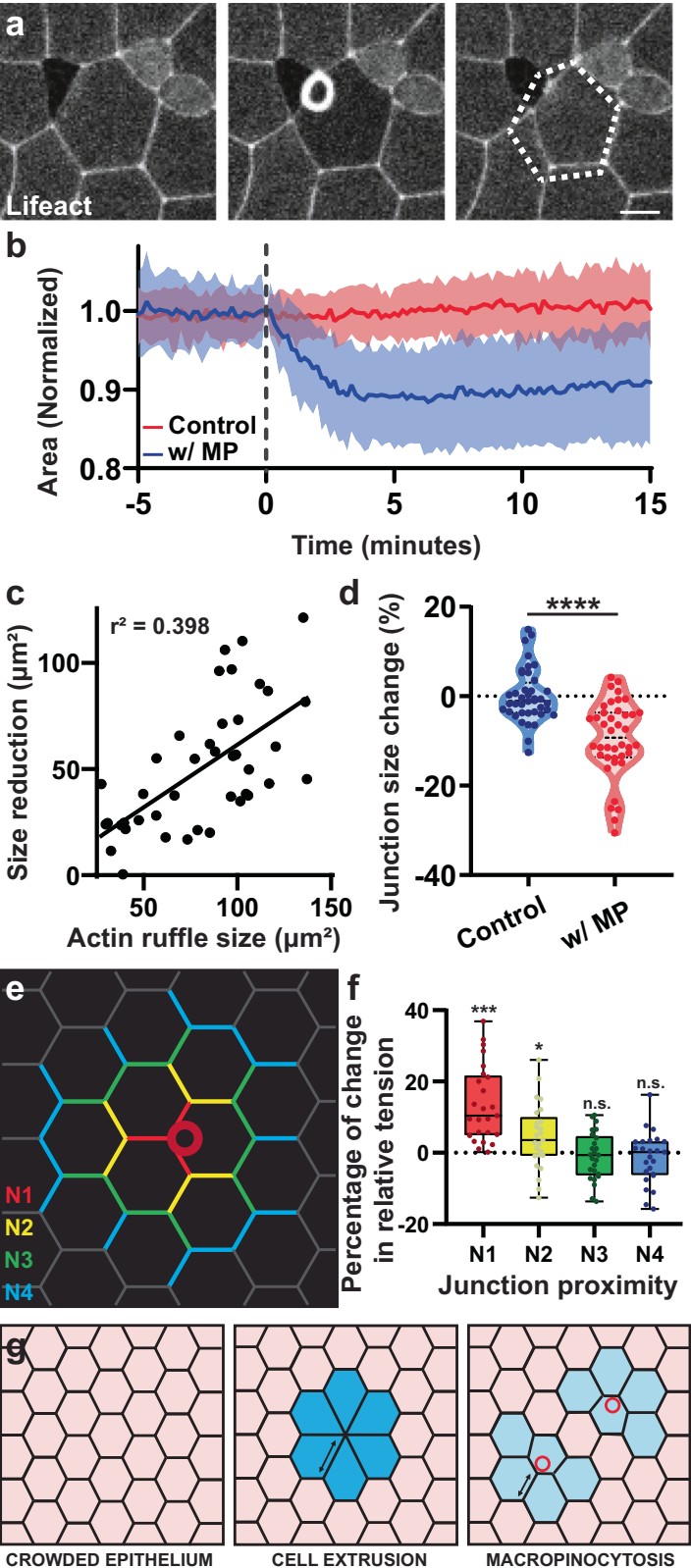

quantifiable bias for them to occur in cells adjacent to MCCs (Fig. 4c–e and Supplementary Movie 1). We hypothesize that crowding-driven MCC extrusion could be affected by apical size reduction of neighboring cells via macropinocytosis.

First, to understand why macropinocytosis is increased around MCCs we mathematically modeled the ciliated epithelium in order to explore how MCCs affect their neighboring cells. For the model we

defined MCCs as non-dividing cells with high stiffness (low deformability) due to the elaborate apical cytoskeleton network they contain to support the mechanical force associated with ciliary beating[38–41]. Our simulations begin with proliferation without MCCs, followed by the introduction of evenly spaced MCCs with the goal of understanding the consequence of these stiffer cells on their neighbors (Fig. 4f, h, j, Supplementary Fig. 2a, and Supplementary Movie 8). Our

**Fig. 3 | Macropinocytosis leads to cell size reduction. a** Time lapse of a macropinocytotic event in tissue labeled with Lifeact-FP. The dotted line represents cell original size, highlighting the cell size reduction. Scale bar 10 μm. **b** Quantification of cell size change in cells with (blue) or without (red) a macropinocytotic event (MP) using cells synchronized to the onset of macropinocytosis and normalized to original cell size (*n* = 3 exp. and 43 cells total; mean and SD). **c** Correlation between the apical size reduction and the size of actin ruffle (*n* = 3 exp. and 43 cells). **d** Quantification of the change in junction length in cells with and without macropinocytosis (*n* = 4 exp. and 40 cells; two-sided unpaired *t*-test, *p*-value = 1.23e-07). **e** Graphical depiction of a cell with different junction types labeled (N1-N4) in different colors based on their distance from the macropinocytotic event. **f** CellFit analysis of time lapse movies quantifying the change in relative junctional tension before and after macropinocytosis (*n* = 25 events; see Fig. S1f; box = interquartile, center line = median, error bars = min/max; two-sided paired *t*-test of before and after the event *p*-value N1 = 0.0001, N2 = 0.0281, N3 = 0.4890, N4 = 0.3492). **g** Model comparing the effect of cell extrusion (center) or macropinocytosis (right) occurring in the cell (asterisk) of a crowded epithelium (left). Blue color highlights the putative effect on cells/neighbors. Source data are provided as a Source Data file.

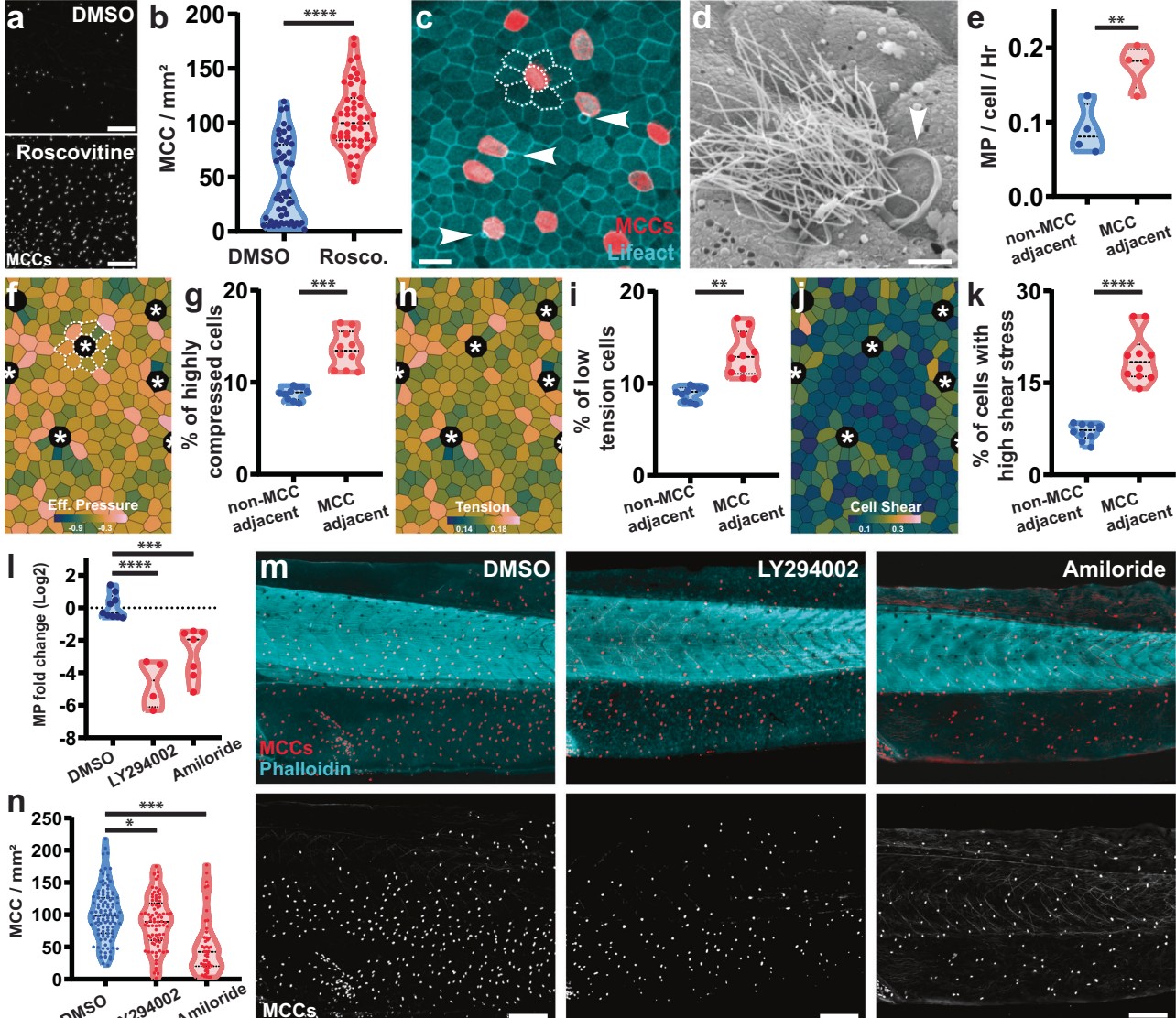

**Fig. 4 | Macropinocytosis regulates tissue crowding during epithelial development. a** Representative images of multiciliated cells (MCCs, marked with acetylated tubulin ab) in ST47 embryo after treatment with DMSO or Roscovitine. Scale bar 300 μm. **b** Quantification of MCC number in embryo treated for 24 h with either DMSO or Roscovitine (*n* = 4 exp. and 50 DMSO and 51 Roscovitine embryos; two-sided unpaired *t*-test, *p*-value = 1.42e-014). **c** Representative image of macropinocytic events (arrowheads) occurring next to MCCs in the epithelium of a TubA1A-membrane RFP transgenic embryo labeling multiciliated cells (red) and injected with Lifeact-FP (cyan). Dashed lines represent rosette-like structure around MCCs. Scale bar 30 μm. **d** Representative SEM image showing a macropinocytotic event (arrowhead) adjacent to an MCC. Scale bar 5 μm. **e** Quantification of the number of events (MP) per cell per hour for cells adjacent or not adjacent to MCCs (*n* = 4 exp. 191 macropinocytotic events; two-sided paired *t*-test, *p*-value = 0.0034). **f, h, j** Images of a simulated epithelium with multiciliated cells (asterisk). Cells are color-coded for effective pressure (**f**), tension (**h**) or shear stress (**j**). Dashed lines represent rosette-like structure around MCCs. **g, i, k** Percentage of non-MCC neighbors and MCC neighbors that are in the highest decile of effective pressure (**g**) and shear stress (**k**) or in the lowest decile of tension (**i**) (*n* = 10 simulations; two-sided paired *t*-test; effective pressure *p*-value = 0.0004, tension *p*-value = 0.0021, shear stress *p*-value = 5.4e-05). **l** Quantification of the fold difference in macropinocytotic events comparing 1 h before and 1 h after treatment with DMSO (*n* = 9 exp.), 5 μM LY294002 (*n* = 4 exp.; *p*-value = 8.0e-06), or 1 mM Amiloride (*n* = 7 exp.; *p*-value = 0.0002); two-sided unpaired *t*-tests. **m** Representative images of MCCs (marked with acetylated tubulin ab; red/white) in ST46 embryos stained with phalloidin (cyan) after treatment with DMSO, LY294002 or Amiloride. Scale bar 300 μm. **n** Quantification of the number of MCCs after treatment with DMSO (*n* = 115 embryos), 5 μM LY294002 (*n* = 81 embryos; *p*-value = 0.0457) or 1 mM Amiloride (*n* = 49 embryos; *p*-value < 0.0001); two-sided unpaired *t*-tests. Source data are provided as a Source Data file.

simulations revealed that the presence of MCCs induce the organization of the surrounding cells into rosette-like structures (Fig. 4f and Supplementary Movie 8), resembling those observed in vivo (Fig. 4c). Additionally, our model predicts that cells in these rosettes are subjected to elevated mechanical stress. Comparing rosette cells, adjacent to MCCs, with non-MCC-adjacent cells we find an enrichment in cells that are highly compressed as revealed by quantifying the percentage of cells that fall in the highest decile of negative effective pressure (Fig. 4f, g and Supplementary Movie 8) and shear stress (Fig. 4j, k and Supplementary Movie 8). Importantly, this population is also enriched in cells that fall within the bottom decile for tension (Fig. 4h, i and Supplementary Movie 8). This local enrichment of compressed, low-tension cells adjacent to MCCs, together with our in vivo observation of increased macropinocytosis around MCCs, supports a model in which MCCs locally compress their neighbors leading to macropinocytosis. These findings reinforce our conclusion that mechanical compression induces macropinocytosis. In addition to the highly compressed low-tension cells, the simulation also drives an increase in cells at the other end of the spectrum, with low compression and high tension which we would predict to be agnostic or even resistant to macropinocytosis (Supplementary Fig. 2b and Supplementary Movie 8).

To properly investigate how local macropinocytosis around MCCs could affect the timing of their extrusion, we utilized two known small molecule inhibitors of macropinocytosis, Amiloride and the PI3K inhibitor LY294002[42,43]. We found that, consistent with the literature, both drugs significantly reduced macropinocytosis in our system (Fig. 4l). Furthermore, extended treatment with either compound leads to premature extrusion of MCCs from the epithelium (Fig. 4m, n) suggesting that, in the absence of macropinocytosis, tissue crowding is accelerated requiring a concomitant increase in cell extrusion. These results suggest that apical size reduction via macropinocytosis can alleviate tissue crowding which, as evidenced by the modulation of MCC extrusion, can have a profound impact on overall tissue architecture.

### External compression triggers a wave of macropinocytosis

While the compressive stress of tissue crowding during development arises from proliferation, externally applied physical forces can also alter the compressive stress experienced by a tissue. In fact, epithelial compression has been reported to result in lower membrane tension[2]. This led us to investigate if macropinocytosis could also provide a response to compression-induced stress. To investigate this, we first embedded embryos in agarose gel, similar to what has been used to induce compression in zebrafish embryos[44]. In embryos compressed by embedding in 2% agarose, we find a rapidly induced wave of macropinocytosis across the epithelium with almost every cell having an event (Supplementary Fig. 3a, b). However, in this rigidly confined space, the embryos do not survive for extended periods of time. We therefore developed a second approach for compression where we pressed embryos using a coverslip within a defined imaging chamber. This led to the flattening (and compression) of only the round parts of the embryo (e.g., the belly), which allowed for local imaging of just the compressed tissue without the lethality of whole embryo agarose compression (Supplementary Fig. 3c–e). This compression induced a similar wave of macropinocytosis (Fig. 5a, b, Supplementary Fig. 3f, and Supplementary Movie 9) that could be imaged for extended periods of time.

Because most cells contained a macropinocytotic event during this wave of macropinocytosis, we observed drastic tissue remodeling as a result of both apical size reduction in the host cell and stretching in the neighboring cells. Quantification of apical size on isolated cells confirmed that macropinocytosis leads to a reduction in the apical surface area, similar to what we observed during epithelial growth (Supplementary Fig. 3g). To evaluate overall tissue remodeling by the wave of macropinocytotic events, we measured absolute changes in apical size that considers both apical size reduction by macropinocytosis as well as cell stretching induced by events in the neighboring cells (Fig. 5c–e and Supplementary Movie 9). We measured a peak in tissue remodeling that corresponds with the timing of the wave of macropinocytotic events (Fig. 5e). Additionally, in embryos in which we have inhibited macropinocytosis, we see a dramatic inhibition of epithelial remodeling via apical size change (Fig. 5f–h). These results indicate that macropinocytosis can serve as an important regulator of tissue remodeling not only during development but also in response to external stimuli.

As mentioned above, cell extrusion is an important response to tissue crowding and is known to occur in response to external compression[4]. As expected, we observed a significant increase in cell extrusion events in response to embryo compression compared to uncompressed embryos (Fig. 5i, j and Supplementary Movie 10). Interestingly, the former neighbors of recently extruded cells have lower rates of macropinocytosis (Supplementary Fig. 3h), most likely because local stretching induced by cell extrusion increases tension and prevents the need for macropinocytosis. Importantly, embryos pretreated for 30 min with the macropinocytosis inhibitors exhibit a further increase in extrusion in response to compression (Fig. 5i). These results indicate that tissue remodeling associated with the wave of macropinocytotic events helps to alleviate the effects of tissue compression. Importantly, in the inhibitor-treated embryos, an increase in cell extrusion compensates for the lack of macropinocytosis-induced apical size reduction. Overall, we propose that apical size reduction by macropinocytosis cooperates with cell removal by extrusion to counter developmental- or environmental-induced compression.

## Discussion

We report a previously unidentified role for macropinocytosis as a regulator of compression in vertebrate epithelia. Maintaining homeostasis requires a constant balance between compression and stretching. An important response to compression is live cell extrusion, which locally stretches the apical area of the tissue. We propose that apical membrane internalization through numerous macropinocytotic events can serve a similar function of apical stretching by causing a cumulative loss of approximately 10% of the apical cell area from multiple cells, rather than the 100% loss of one cell during extrusion.

We observe this mechanism during development, where proliferation leads to compression in the form of tissue crowding. We propose that a decrease in membrane tension induced by crowding triggers macropinocytotic events and that cell size reduction by macropinocytosis reshapes tissue architecture to alleviate crowding. The long-term importance of this mechanism for epithelial regulation is evidenced by the increase in cell extrusion when macropinocytosis is inhibited. Importantly, we show that this role for macropinocytosis is versatile as it is also utilized in acute response to external compression where a wave of macropinocytotic events extensively remodel the epithelium to alleviate the effects of compression. During external compression, there is an increase in cell extrusion that is significantly amplified by the inhibition of macropinocytosis. We therefore propose that macropinocytosis can act as a rapid first response to compression that serves to maintain tissue integrity and minimize cell loss. When macropinocytosis fails or proves insufficient, the epithelium relies on the more extreme and irreversible cell extrusion.

This regulatory mechanism could also be at play in other vertebrates, including humans. Macropinocytosis and circular actin ruffling are widely conserved mechanisms reported in numerous human tissues. For example, similar actin-based internalization events have recently been reported in intestinal organoids[45]. Additionally, macropinocytosis has been observed in evolutionarily diverse organisms,

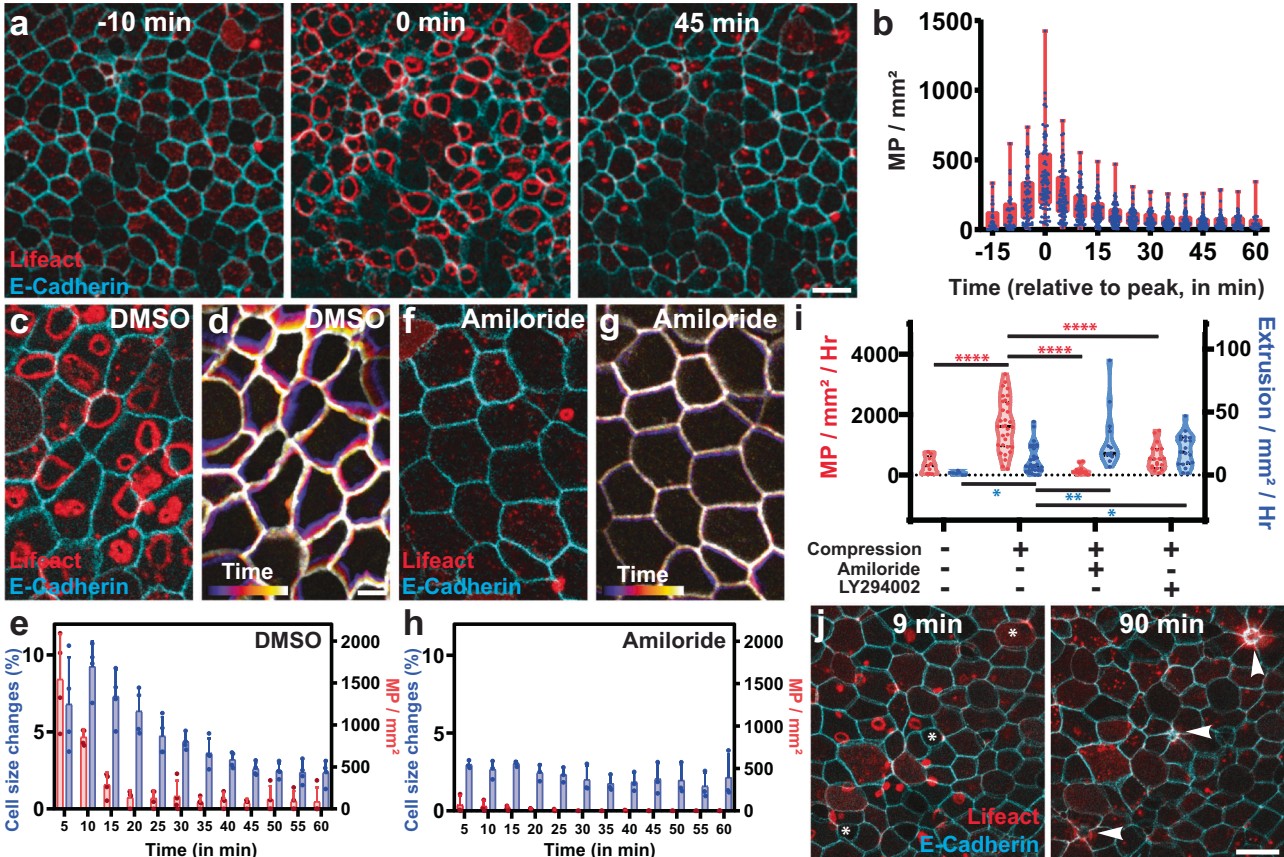

**Fig. 5 | A wave of macropinocytosis remodels the epithelium in response to external tissue compression. a** Representative image of an embryo injected with Cadherin-FP (cyan) and Lifeact-FP (red) before (left), during (middle), and after (right) a wave of macropinocytosis induced by tissue compression. Scale bar 30 μm. **b** Quantification of the wave of macropinocytotic events (MP) upon compression (*n* = 80 scan areas from 32 embryos from 12 exp.; bars = mean, error bars = SD). **c, f** Representative images of embryos injected with Cadherin-FP (cyan/white) and Lifeact-FP (red) 5 min after compression in control (**c**) or Amiloride treated (**f**) embryos. Scale bar 10 μm. **d, g** Representation of tissue remodeling over 10 min using a temporal color code of control (**d**) or Amiloride (**g**) treated embryos. Scale bar 10 μm. **e, h** Quantification of tissue remodeling (i.e., overall change in cell apical area)

(blue; left axis) and macropinocytotic events (red; right axis) in DMSO (**e**) and Amiloride treated (**h**) embryos upon compression (*n* = 3 exp.; bar = mean, error bars = SD). **i** Quantification of macropinocytotic events (red; left axis) and cell extrusion events (blue; right axis) in non-compressed (*n* = 9 embryos; MP *p*-value 5.00e-05, extrusion *p*-value = 0.0168) and compressed embryos in either DMSO (*n* = 32 embryos) or treated with amiloride (*n* = 13 embryos; two-sided unpaired *t*-tests; MP *p*-value = 1.31e-07, extrusion *p*-value = 0.0028) or LY294002 (*n* = 19 embryos; MP *p*-value = 4.19e-06, extrusion *p*-value = 0.0272). **j** Representative image of an embryo injected with Cadherin-FP (cyan) and Lifeact-FP (red) having both macropinocytotic events and cell extrusion (arrowhead) upon compression. Scale bar 30 μm. Source data are provided as a Source Data file.

from the marine organism Hydra to human cells in culture[13,23]. Finally, our findings with macropinocytosis add to the growing list of diverse functions for endocytic processes[9,10,12,46]. Emerging evidence supports the broad potential use of macropinocytosis as a rapid and reversible response to compressive stresses. Additionally, the removal of apical membrane in response to decreased membrane tension could act as a feedback mechanism to restore tension, suggesting a potential role for macropinocytosis in regulating apical membrane tension. Ciliated epithelia are notable for the large amount of exocytosis required to deliver mucus to the apical surface. This exocytosis will bring excess membrane to the apical surface, which should drive a decrease in tension. Macropinocytosis could therefore also serve as a homeostatic mechanism to counteract this drift toward decreased tension, maintaining epithelial integrity by balancing membrane addition with membrane removal.

Overall, this role for macropinocytosis offers an additional mechanism for vertebrate epithelia to regulate tissue crowding, particularly on relatively short timescales, and to continuously adjust tension distribution in response to stresses from both the environment and morphogenic growth. Our findings enhance the understanding of epithelial dynamics, highlighting how constant tissue remodeling is central to stress response.

## Methods

### Transgenic Xenopus and mRNA embryo injections

In vitro fertilizations were performed using standard protocols[47–50] that have been approved by the Northwestern University Institutional Animal Care and Use Committee (IS00027128). Transgenic Xenopus expressing membrane-bound RFP driven by the tubulin promoter (Xla.Tg(tuba1a:MyrPalm-mRFP)^NXR), were previously generated and obtained from the National Xenopus Resource Center (NXR)[51]. Embryos were used between fertilization and stage 48. Sex was not considered as a biological variable because Xenopus embryos at these stages have not yet undergone sexual differentiation. Therefore, data were not disaggregated by sex. Wild type or transgenic embryos were injected at the two- or four-cell stage with 50 pg mRNA. mRNA of Lifeact-GFP/RFP[40], ZO-1-RFP[52], E-Cadherin-GFP[52], SF9-GFP[16], membrane-RFP[53] were synthesized with the Sp6 mMessage Machine kit (Life Technologies, AM1340) and purified by RNeasy MiniElute Cleanup Kit (QIAGEN, 74204).

### Immunostaining

Embryos were fixed with 4% PFA/PBS. To visualize cilia, embryos were incubated with mouse anti acetylated α-tubulin (T7451; Sigma-Aldrich, 1:500) in 0.1%Triton/PBS for an hour at room temperature followed by

Cy-2-conjugated goat anti-mouse secondary antibodies (Thermo Fisher Scientific) at 1:750 dilution in 0.1%Triton/PBS for two hours at room temperature. Phalloidin 568 (A12380, Invitrogen, 1:750) was used to visualize actin. Cell density was quantified manually using the FIJI software.

## Microscopy

Fluorescent light imaging was performed either using a Nikon A1R laser scanning confocal microscope (Figs. 1a, b, d–f, h–l, 2a–c, e–h, 3a–d, f, 4c, e, l, and 5a–j; Supplementary Figs. 1a–f, 3a, b, f–h, and Supplementary Movies 1, 2, 3, 4, 5, 6, 7, 9, and 10) or a Nikon Ti2 microscope used as a widefield microscope (Fig. 4a, b, m, n and Supplementary Fig. 3c–e) or using a MizarTILT light sheet (Figs. 1g, 2h, and 4e, l) and a photometric Prime 95B camera. Objectives used included Plan Fluor 10× Ph1 DLL (NA 0.3), Plan Fluor 20× MImm DIC N2 (NA 0.75), Plan Fluor 40× Oil DIC H N2 (NA 1.3), Plan Apo VC 60× Oil DIC N2 (NA 1.4), and SR HP Apo TIRF 100 × H (NA 1.49). Live embryos were imaged in anesthetic (0.01% Tricaine in 0.1× MMR).

Quantifications were done using FIJI Software[54]. Cell density was quantified manually using the FIJI software. Cell size change tracking was performed using the FIJI TrackMate plugin coupled with Cellpose segmentation[55] on embryos co-expressing Cadherin-GFP for segmentation and Lifeact-RFP for ruffles visualization. Junction sizes were quantified manually using FIJI. To visualize dextran internalization, embryos expressing Lifeact-GFP were imaged in anesthetic with 0.125 μg/ml of 70 kDa TRITC-dextran (Chondrex). Quantification was performed using the linescan tool of FIJI. Quantification of adhesion protein internalization was performed on embryos expressing either Lifeact-RFP and E-Cadherin-GFP or Lifeact-GFP and ZO-1-RFP using the linescan tool of FIJI. Quantification of active myosin around the ruffles was performed on embryos expressing Lifeact-RFP and SF9-3 × GFP using the linescan tool of FIJI. To quantify preferential macropinocytosis around multiciliated cell ruffles, ruffles were manually counted and classified based on whether they occurred in MCC-neighboring or non-MCC-neighboring goblet cells, and then divided by the total number of MCC-neighboring or non-MCC-neighboring goblet cells. Quantification of MCC density was performed on z-stacks in FIJI using the DoG detector of the TrackMate plugin from FIJI.

For SEM preparation, embryos were fixed for at least 24 h in 2% glutaraldehyde on 0.1 M sodium cacodylate buffer at 4 °C overnight and postfixed using 1% OsO4 in water for 2 h. The specimens were then dehydrated in a graded series of ethanol (50%, 70%, 90%, and 100%), critical-point-dried with carbon dioxide (Samdri-790, Tousimis, Rockville, USA), mounted, and coated with 10 or 20 nm gold in a sputter coater. Finally, the specimens were observed under a scanning electron microscope (JCM- 6000PLUS, JEOL, Japan).

## Drug treatments

To assess the effect of small molecules on the level of macropinocytosis, embryos expressing Lifeact-FP were first imaged for at least 30 min in anesthetic to establish their basal level of macropinocytosis. Subsequently, they were imaged for at least 45 min in anesthetic supplemented with either $^1/_{1000}$ DMSO, 1 mM Amiloride (#14409, Cayman Chemical), 2 μM LY294002 (#HY-10108, MedChemExpress), 20 μM Blebbistatin (#B0560, Sigma), 0.25 mM Sodium Deoxycholate (#D6750, Sigma), 2 μM GsMTx4 (#HY-P1410, MedChemExpress), or 10 μM Yoda1 (#558610, Fisher). For cyclodextrin treatment, embryos were incubated for one hour in 50 mM Methyl-β-cyclodextrin (#J66847, Thermo) diluted in 0.1× MMR before being imaged for another hour in anesthetic. The level of macropinocytosis was manually quantified using FIJI, and results were expressed as $\log_2(^{treatment\ level}/_{basal\ level})$. Effective internalization upon drug treatment was assessed in drug-treated embryos expressing Lifeact-GFP, imaged in anesthetic with 0.125 μg/ml of 70KDa TRITC-dextran. To assess the effect of macropinocytosis on MCCs extrusion, embryos were treated from stage 43 with either DMSO, 1 mM Amiloride, or 2 μM LY294002 in 0.1× MMR. Media were changed after 24 h, and embryos were fixed in 4% PFA/PBS at 48 h of treatment before processing for immunostaining.

To assess the effect of proliferation on goblet cell density, macropinocytosis or MCC extrusion, embryos were treated for 24 h from stage 44/45 with either DMSO or 50 μM Roscovitine (#HY-30237, MedChemExpress) in 0.1× MMR. Subsequently, embryos were either imaged for 30 min to measure macropinocytosis level or fixed and processed for immunostaining to assess goblet cell density or MCCs extrusion.

## Tension inference with CellFIT

To infer junction tension, embryos co-expressing Lifeact-RFP to visualize ruffle events and Cadherin-GFP for tissue segmentation were used. Tissue segmentation was performed using TissueAnalyzer with additional manual correction[56]. The hand corrected segmented images were processed with Zazu CellFIT[35] to obtain inferred tension. The results from 1 to 4 images before and after macropinocytosis events were averaged to obtain the final tension inference by junction category (N1, N2, N3, N4), expressed as a ratio ($^{average\ tension\ after\ macropinocytosis}/_{average\ tension\ before\ macropinocytosis}$).

## Compression assay

To assess the effect of agarose compression on macropinocytosis levels, embryos expressing Lifeact-FP were imaged in 2% agarose gel prepared in 0.5× anesthetic. For compression induced by coverslip, embryos under anesthesia were placed on a glass-bottom imaging dish (#P35G-1.5-14-C, Mattek), with a circle of grease applied outside the glass-bottom, and an 18 mm coverslip (#72222-01, Electron Microscopy Science) placed on top of the grease circle. To assess the extent of compression, embryos were imaged before compression, then compressed, fixed, and re-imaged after removing the compressing coverslip (Supplementary Fig. 3c–e). To examine the effect of macropinocytosis inhibition on compression-induced tissue remodeling, embryos expressing Lifeact-FP and Ecad-FP were pretreated with either $^1/_{1000}$ DMSO or 1 mM Amiloride in 0.1× MMR for 30 min before imaging for 1 h under coverslip compression. Cell size quantification was performed using the FIJI Trackmate plugin coupled with cellpose segmentation[55], while actin ruffles were manually quantified using FIJI software. To assess the impact of macropinocytosis inhibition on compression-induced cell extrusion, embryos expressing Lifeact-FP were pretreated with either $^1/_{1000}$ DMSO, 1 mM Amiloride, or 2 μM LY294002 in 0.1× MMR for 30 min before imaging for 1 h under coverslip compression. Manual quantification of embryo height, macropinocytosis or cell extrusion was conducted using FIJI software. Macropinocytosis levels around extruded cells were quantified within 10 min of extrusion, comparing direct neighbors of the extruded cells to cells within a radius twice the neighbor area. Effective internalization upon coverslip compression was assessed in embryos expressing Lifeact-GFP, imaged in anesthetic with 0.125 μg/ml of 70KDa TRITC-dextran.

## Ectodermal explants

To visualize macropinocytosis in explants and caps (organoids), animal caps from embryos expressing Lifeact-FP were dissected at stage 10 (ST10) and processed following methods described previously[40]. Briefly, explants were either allowed to round up and cultured in 0.5× MMR or plated between a fibronectin-coated coverslip and an imaging glass-bottom dish (#P35G-1.5-14-C, Mattek), cultured in 0.5× DFA supplemented with BSA. Uncapped embryos served as a stage reference. Round-up and adherent explants were imaged at stage 34.

## Tissue simulations

Simulations were performed using the vertex model code developed in the authors' previous work[57]. This method uses incidence matrices to specify the topology of the vertex network[58,59], allowing for direct manipulation and formulation of monolayer mechanics, and employing logarithmic relationships between forces and strains. Within the context of a vertex model simulation of an isolated epithelial monolayer, MCCs were modelled by focusing on their increased stiffness and lack of cell division. MCCs are distinguished from other cells by a prefactor, $\sigma$, multiplying the cell cortical tension and internal pressure. This means that a given difference between cell perimeter length and preferred perimeter length produces a tension force that is greater by a multiple equal to this prefactor for MCCs than it would be for bulk cells. Similarly, a given difference between cell area and preferred area gives a pressure force in MCCs greater than for bulk cells by a factor of $\sigma$. MCCs are also prevented from dividing, whereas bulk cells can divide when their age exceeds the cell cycle time. Cells were chosen to be MCCs by randomly selecting an initial cell within the monolayer, excluding cells at the monolayer periphery. This cell was made an MCC, and its neighbours, all of its neighbours' neighbours, and all of their neighbours were removed from the pool of cells that could be chosen to be MCCs, ensuring that no path of fewer than 3 adjacent cells can be formed between any 2 MCCs. Another cell was then selected from the remaining pool of bulk cells, again excluding peripheral cells and those excluded by proximity to an existing MCC, and the process was repeated until there were no remaining bulk cells that had not been excluded.

For the data in this work, we produced 10 independent simulations for each parameter set. Monolayers were grown from an initial regular geometry of 61 cells to a size of 400 cells. At this point, a selection of cells was randomly chosen following the protocol described previously, and these cells were made MCCs. The stiffness of these MCCs was changed by a factor of $\sigma$, and they were no longer allowed to divide. The simulation was allowed to continue until the monolayer had grown to 800 cells, at which point all division was stopped and the monolayer was allowed to relax to equilibrium.

We ran simulations with $\sigma = 1$, meaning MCCs are the same stiffness as bulk cells but do not divide, and $\sigma = 10$. All of these simulations were repeated whilst allowing MCCs to divide, such that for $\sigma = 1$, MCCs are entirely indistinguishable from bulk cells. For analyses, all cells at the periphery of the monolayer were excluded. All simulations were performed in the jammed regime with preferred perimeter $L_0 = 0.75$ and relative stiffness $\Gamma = 0.2$, with uniform external pressure (0.5).

## Statistics & reproducibility

No statistical method was used to predetermine sample size. No data were excluded from the analysis. The experiments were not randomized, and the investigators were not blinded to allocation during experiments or outcome assessments. When possible, internal controls were included to ensure data consistency and experimental reliability. All key findings were reproduced in at least three independent experiments using embryos from different clutches. Statistical analysis was performed as described in the figure legends.

## Reporting summary

Further information on research design is available in the Nature Portfolio Reporting Summary linked to this article.

## Data availability

Source data supporting the findings of this study are provided with this paper. Due to the size of our imaging sets it is not currently practical to download all images. Specific data are available from the corresponding author upon request. Source data are provided with this paper.

## Code availability

Code and analysis scripts can be found in the MCC branch of the GitHub repository (https://github.com/chris-revell/VertexModel/tree/MCC-branch)[60] or (https://doi.org/10.5281/zenodo.15424908).

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

## Acknowledgements

This work was supported by the Wellcome Trust (225408/Z/22/Z to Sarah Woolner), the Biotechnology and Biological Sciences Research Council (BB/T001984/1 to Oliver Jensen), The Leverhulme Trust (RPG-2021-394 to Sarah Woolner and Oliver Jensen), and NIH-NIGMS (R01GM089970 to Brian Mitchell). We would like to thank the National Xenopus Resource center and Xenbase for their important community service that was critical for this work. Some of the Imaging was performed at the Northwestern University Center for Advanced Microscopy (RRID: SCR_020996) generously supported by CCSG P30 CA060553 awarded to the Robert H Lurie Comprehensive Cancer Center.

## Author contributions

Conceptualization: E.B. and B.J.M. Methodology: E.B., C.R., O.J. and B.J.M. Investigation: E.B., E.E.S., C.R., O.A.H., A.G., J.S., J.M. and F.K. Software: CR and OJ. Visualization: E.B., C.R. and B.J.M. Funding acquisition: S.W., O.J. and B.J.M. Project administration: B.J.M. Supervision: C.A., S.W., O.J. and B.J.M. Writing–original draft: E.B. and B.J.M. Writing–review & editing: E.B., C.R., E.E.S., A.G., J.S., J.M., S.W., O.J. and B.J.M.

## Competing interests

The authors declare no competing interests.

## Additional information

**Supplementary information** The online version contains
supplementary material available at

Brian Mitchell.

**Peer review information** *Nature Communications* thanks Adam Hoppe,
and the other, anonymous, reviewer(s) for their contribution to the peer
review of this work. A peer review file is available.

