## [Transparent Peer Review file · Nature Communications]

Apical Size Reduction by Macropinocytosis Alleviates Tissue Crowding

Corresponding Author: Dr Brian Mitchell

Version 0:

Reviewer comments:

Reviewer #1

(Remarks to the Author)

The manuscript by Bresteau et al. investigates a newly identified mechanism by which epithelial tissues manage crowding and respond to mechanical stresses, focusing on the role of macropinocytosis. This process, involving the internalization of portions of the apical membrane, provides a non-destructive alternative to live cell extrusion for alleviating crowding and maintaining tissue integrity. Using the *Xenopus* embryonic epidermis as a model system, the study reveals how macropinocytosis facilitates dynamic tissue remodeling during development and in response to external forces. The study documents frequent macropinocytotic events in developing epithelial tissues, characterized by actin ruffling, where circular protrusions on the cell surface internalize apical membrane regions. These events often occur near cell-cell junctions and become increasingly prevalent as tissue density grows during development, indicating that crowding triggers macropinocytosis.

The authors explore the regulatory mechanisms underlying macropinocytosis, focusing on the role of membrane tension and mechanosensation. They find that reduced membrane tension in crowded or compressed tissues promotes macropinocytotic activity, while higher tension suppresses it. Pharmacological manipulation of tension and mechanosensory pathways further supports the role of these factors in modulating macropinocytosis. The authors propose that macropinocytosis is a direct response to crowding-induced decreases in membrane tension, functioning to alleviate compression by reducing the apical size of individual cells.

Detailed imaging and quantitative analysis show that macropinocytosis reduces the apical surface area of cells by approximately 10%, alleviating crowding and redistributing local mechanical tension across the tissue. This process facilitates broader tissue remodeling and minimizes the need for cell extrusion. When macropinocytosis is inhibited, cell extrusion increases significantly, indicating that the two mechanisms work in tandem to manage tissue stress.

Macropinocytosis appears to act as a primary, non-destructive response, reducing reliance on the irreversible process of cell extrusion.

The study also investigates how macropinocytosis responds to external compressive forces. External compression triggers waves of macropinocytotic events across epithelial tissues, leading to extensive remodeling as cells reduce their apical size and redistribute tension among neighboring cells. Inhibiting macropinocytosis under these conditions amplifies cell extrusion, further emphasizing its critical role in maintaining tissue structure during mechanical stress.

The role of macropinocytosis is further explored in the context of multiciliated cells (MCCs), specialized epithelial cells. Macropinocytotic events are found to occur preferentially in cells neighboring MCCs, potentially influencing the timing of MCC extrusion. Through mathematical modeling, the authors demonstrate that MCCs create localized mechanical stresses due to their rigidity and inability to divide, which promotes macropinocytosis in adjacent cells. This dynamic highlights the process's role in maintaining epithelial architecture.

This study contributes significantly to the understanding of how epithelial tissues adapt to developmental and mechanical pressures. It identifies macropinocytosis as a key mechanism for managing crowding and compression, offering a reversible and non-destructive means of tissue remodeling. The findings have broad implications, as macropinocytosis and actin-driven processes are evolutionarily conserved and relevant in other species, including humans. This work sets the stage for future studies to explore the roles of macropinocytosis in other biological systems and its implications for conditions like tumor progression and epithelial disorders.

While the results presented are intriguing, it would be important for the authors to clarify certain aspects of their study. Specifically, it remains unclear whether the discussion pertains to membrane tension, cortical tension, or both. These are distinct concepts, and their interchangeable use in the manuscript creates ambiguity. For example, experiments altering lipid content would primarily affect membrane tension, whereas treatments with blebbistatin would largely influence cortical

tension. Clear differentiation between these terms and their respective impacts would enhance the interpretation of the findings.

Additionally, the type of strain being described warrants further clarification. The assertion that *Xenopus* organoids exhibit low tension while flat explants exhibit high tension requires more precise terminology. "Tension" may not accurately describe the mechanical state in this context, and the manuscript would benefit from a clearer explanation of the mechanical forces at play.

Finally, the relationship between mechanical pressure applied to the tissue and tissue crowding requires further elaboration. It is unclear how pressing on the epithelium is mechanistically linked to tissue crowding. One might expect lateral compression of the epithelium to lead to crowding, whereas vertical compression from above could trigger mechanosensory responses unrelated to increases in cell density. Could the authors clarify these points and elaborate on how these forces contribute to their findings? Addressing these issues will strengthen the overall clarity and impact of the manuscript.

Reviewer #2

(Remarks to the Author)

I have read, "Apical Size Reduction by Macropinocytosis Alleviates Tissue Crowding" By Bresteau et al. The authors describe, using beautiful microscopy methods, a novel role for macropinocytosis in the removal of apical plasma membrane from *Xenopus* embryo epithelium in response to the compression of the apical plane. Major revision of the article aimed at clarifying the interpretation of the results should make these findings will be of broad interest and impact in the field of cell biology. This is an excellent study. Clarification of its interpretation will make it more accessible.

The main concern I have is that the authors claim that macropinocytosis is 'alleviating crowding by reducing cell size.' It is entirely unclear how the formation of a large (micrometers in diameter) endosome would reduce cell size. The lack of precision in the biophysical description needs improvement. Two major points that need clarification:

1) While internalization 10% of the apical membrane into a vesicle may reduce the area of the apical membrane it should increase the cell volume (size). If 'size' herein refers to the apical area then maybe say 'apical area.'

2) Rather than alleviate crowding, these data suggest that macropinocytosis is a mechanism to maintain apical/junctional tension by removing membrane. This explanation seems a better fit with the data including the inhibitor data -> Yoda1 (simulating extreme tension, halts macropinocytosis) and GSTMx – simulating no tension increases macropinocytosis. Moreover, compression of the embryos crowding the apical plane drives intensive macropinocytosis => restoring apical tension by reduction of apical surface area (10%!), while costing an infinitesimal change in cell volume (I would guess on the order of (<1%)). Some discussion of this would be helpful).

3) Move S9 and figure 5J appear to cooperate the notion that macropinocytosis is maintaining tension on the apical surface. Regions where cells are extruded appear to have low rates of macropinocytosis (suggesting elevated/sufficient tension), whereas regions distal to extrusion events appear very active. Could the frequency of macropinosomes relative to extrusion sites be quantified?

4) There is discrepancy between measuring the number of macropinosomes and ruffles. The switch to ruffles/hr/mm² vs macropinosomes/hr/mm² is perplexing – and has a different implication – the membrane is not removed in a ruffle. The distinctive shape, supported by the dextran uptake, is enough for this reviewer to generalize to just calling them macropinosomes.

Figure 2H. has a y-axis labeled "Ruffles fold change (log 2)" The legend says that this is the "quantification of the fold difference in macropinocytosis level..." This is quite a perplexing figure. Is it measuring ruffles or macropinosomes? What is meant by fold change (of what)? This is a very important plot and needs clarification and attention. Within this plot does each dot represent an embryo?

I would have expected something like # of macropinosomes/cell/time.

Figure 2H What was measured? How was it measured?

The dextran uptake assay under these inhibitors would be more compelling, but imaging of ruffles should suffice. Data showing examples of ruffle morphology under drug conditions should also be shown.

Figure 3B – what are the halos colors above and below in the line plot? These seem far too small for variance. Individual traces should be shown.

This is greatly helped by plots C and D – which indicate that 3B does not display a meaningful variance.

Figure 3E – "junction types labeled (N1-N4)" ...maybe... junction proximity (or edge-linkage) to the macropinosome? "Types" sounds vague.

Similarly, this statement (top of page 4) doesn't work for me "cell size reduction is primarily driven by internalization of a large portion of the apical membrane rather than internalization of the junctions per se."

The apical area is shrinking. But the volume of the cell would increase owing to the internalization of extracellular fluid within the macropinosome. This seems to be more of a shape change – presumably creating an elongation (possibly infinitesimal) in the axes perpendicular to the plane of the junctions.

Similar comments for the paragraph in the middle of page 4. There is no evidence that the volume of the cell decreases. In fact, the images in 1.H' appears to show f-actin near the basal surface extend downward after the macropinosome forms. 1.I' doesn't seem to show the basal membrane?

Could this be quantified? Granted this is the worst resolution direction and position of the confocal microscope.

Similarly in the 2D explant model, (Figure 2.D, E), part of the created tension would presumably be from the basal side of the cells pressed down against a rigid surface. A cell could only decrease its apical plane size by 1) getting taller (extrusion) or by 2) expelling some its material. Reduction in cell size does not seem accurate. Reduction in apical surface does.

Figure 4. The flip-flopping of color bars makes this figure almost intractable. Why flip the colorbar from low-peach->high-blue for pressure and then invert it for tension? Wouldn't it follow that cells under lateral pressure would have lower mean edge tension? -> same colormap? Shear strain could also be shown a on a similar colormap.

To make figure 4F-K more convincing, the data for MCC-adjacent high-tension cells should also be plotted. It appears that from the images and movies that MCC cells are enriching high and low tension cells. Over MCC-devoid regions.

The stepwise introduction of MCC cells in the simulation should also be described and interpreted in the results section for figure 4. The simulation appears to handle this change well, even though there is a noticeable shift in the compression and shear patterns, when they are introduced.

The methods for compression in agarose gel are unclear – how long were the embryos embedded? Was any external force applied?

For the grease compression – can an estimate be given for how much the embryo's were flattened? E.g. approximately how much were they compressed? This would be important for others to reproduce the work.

“To evaluate overall tissue remodeling by the wave of macropinocytotic events, we measured absolute changes in cell size that considers both cell size reduction by macropinocytosis as well as cell stretching induced by events in the neighboring cells (Fig 5C-E, Movie S8).”

What is meant by absolute changes in cell size? This sounds like area and volume, but I don't think that is what was being measured.

Reviewer #3

(Remarks to the Author)

The manuscript by Bresteau et al describes macropinocytosis in embryonic epithelium during tissue crowding or under compression. The authors show with quantitative imaging of live or fixed samples and SEM that micropinocytosis occurs in normal epithelium in developing embryos. Using chemical regulators of cell division, membrane tension, or tension sensors, they demonstrate that tissue crowding and the resulting changes in tension influence the occurrence of micropinocytosis. Furthermore, they reveal that tissue compression regulates micropinocytosis in embryonic epithelial cells. As micropinocytosis leads to reduction in apical cell size, the authors propose that micropinocytosis is an important mechanism, together with cell extrusion, in tissue response to crowding and compression and contributes to maintenance of epithelial integrity.

The study follows a novel observation of micropinocytosis in *Xenopus* epithelium and investigates its role in tissue remodeling with a series of logical and careful experiments. Multiple manipulations are performed in parallel, quantitative image analysis is provided for each figure, and mathematical simulation is used to model patterns of tension distribution and micropinocytosis events. The data are convincing, and the conclusion is solid.

Some minor concerns are indicated in the following.

In Fig. 4A and B, Roscovitine is used to block cell proliferation/crowding, yet more MCCs are observed. It will be valuable to include cell density measurement together with MCCs here to show that before and after treatment, the number of MCCs do not change, but the loss of MCCs in control embryos is prevented under the treatment condition.

In Fig. 4M, the pattern of MCCs in LY294002-treated embryos seems to be different from that of DMSO- (change the label “Control”) or Amiloride-treated embryos. Is this true? Please provide some explanation about this.

Version 1:

Reviewer comments:

Reviewer #1

(Remarks to the Author)

My concerns have been fully addressed by the authors.

(Remarks on code availability)

Reviewer #2

(Remarks to the Author)

The authors have significantly improved the manuscript and made thorough revisions. I recommend the paper for publication.

(Remarks on code availability)

The URL above appears to be for a specific branch of code that does not open. The VertexModel however, does open. Please check the URL.

The components of the code used in the paper all appear to be in the repository along with readme and license files. I did not install and run the code.

Reviewer #3

(Remarks to the Author)

The authors have addressed all my previous concerns and improved the clarity and quality of the manuscript. The finding has broad impact on our understanding of macropinocytosis in epithelial remodeling during development and in response to mechanical changes. I recommend publication of the manuscript.

(Remarks on code availability)

We thank the reviewers for their supportive comments and their critiques that we believe have led to a stronger manuscript in revision. Below is a point-by-point response to the reviewer comments, but in general we have put considerable effort into clarifying our terminology to specifically describe our results, our experimental approach and our interpretations.

Reviewer #1:

Specifically, it remains unclear whether the discussion pertains to membrane tension, cortical tension, or both. These are distinct concepts, and their interchangeable use in the manuscript creates ambiguity. For example, experiments altering lipid content would primarily affect membrane tension, whereas treatments with blebbistatin would largely influence cortical tension. Clear differentiation between these terms and their respective impacts would enhance the interpretation of the findings.

We apologize for the ambiguity and agree that we could have better defined tension. We have added text to clarify this through the paper to state that we are primarily focused on membrane tension. While we agree with the reviewer that blebbistatin would largely affect cortical tension, in the context of this experiment we were trying to make the point that myosin activity specifically at the actin ruffle was essential for the formation of the macropinocytotic structure.

Additionally, the type of strain being described warrants further clarification. The assertion that *Xenopus* organoids exhibit low tension while flat explants exhibit high tension requires more precise terminology. "Tension" may not accurately describe the mechanical state in this context, and the manuscript would benefit from a clearer explanation of the mechanical forces at play.

We have added text that more precisely describes the explant vs. organoid experiments. Specifically, we make the point that with an underlying ECM substrate explants spread across the surface stretching the cells and leading to lower cell density, which is lost in the 3D organoids that do not have an underlying ECM substrate.

Finally, the relationship between mechanical pressure applied to the tissue and tissue crowding requires further elaboration. It is unclear how pressing on the epithelium is mechanistically linked to tissue crowding. One might expect lateral compression of the epithelium to lead to crowding, whereas vertical compression from above could trigger mechanosensory responses unrelated to increases in cell density. Could the authors clarify these points and elaborate on how these forces contribute to their findings?

We concede that the vertical compression is a blunt tool that is somewhat difficult to interpret. However, the increase in cell extrusion (independent of macropinocytosis) is consistent with our interpretation. Important to note and something we have emphasized in the revision is that our analysis is done on the rounded part of the belly. As a round

tissue becomes flattened, there will be lateral compression of cells towards the middle of the tissue which is where our image analysis is performed.

Reviewer #2:

The main concern I have is that the authors claim that macropinocytosis is 'alleviating crowding by reducing cell size.' It is entirely unclear how the formation of a large (micrometers in diameter) endosome would reduce cell size. The lack of precision in the biophysical description needs improvement. Two major points that need clarification:

1) While internalization 10% of the apical membrane into a vesicle may reduce the area of the apical membrane it should increase the cell volume (size). If 'size' herein refers to the apical area, then maybe say 'apical area.'

We agree with the reviewer that the internalization of a large endosome would likely increase overall cell size. While we thought our previous title made it clear that we were referring to apical cell size, we absolutely appreciate that this was not made clear throughout the paper and have changed the wording accordingly.

2) Rather than alleviate crowding, these data suggest that macropinocytosis is a mechanism to maintain apical/junctional tension by removing membrane. This explanation seems a better fit with the data including the inhibitor data -> Yoda1 (simulating extreme tension, halts macropinocytosis) and GSTMx – simulating no tension increases macropinocytosis. Moreover, compression of the embryos crowding the apical plane drives intensive macropinocytosis => restoring apical tension by reduction of apical surface area (10%!), while costing an infinitesimal change in cell volume (I would guess on the order of (<1%)). Some discussion of this would be helpful).

A role for macropinocytosis as a tension regulator is an exciting idea, especially since it has been shown for other endocytic processes. We have added discussion of this point including the idea that in an exocytosis heavy tissue (such as a mucociliary epithelium) there would be constant need for maintaining apical tension.

3) Move S9 and figure 5J appear to cooperate the notion that macropinocytosis is maintaining tension on the apical surface. Regions where cells are extruded appear to have low rates of macropinocytosis (suggesting elevated/sufficient tension), whereas regions distal to extrusion events appear very active. Could the frequency of macropinosomes relative to extrusion sites be quantified?

We thank the reviewer for spotting this interesting effect, the quantification did confirm that the direct neighbors of extruding cells have lower levels of MP events compared to more distal areas. We have added this new data to Figure S3h.

4) There is discrepancy between measuring the number of macropinosomes and ruffles. The switch to ruffles/hr/mm² vs macropinosomes/hr/mm² is perplexing – and has a different implication – the membrane is not removed in a ruffle. The distinctive shape, supported by the dextran uptake, is enough for this reviewer to generalize to just calling them macropinosomes.

We apologize for this confusing wording / representation. It was based on the fact that for most of our experiments we quantified the actin ring (ruffle) rather than the endocytic event (e.g. marked by fluorescent dextran). While we have not redone all of the experiments, we have validated that in each of our experimental approaches there is a clear internalization event whenever we see an actin ring. This is now shown for the drug treatments in Movie S7 and we have switched all of our figures to quantifying macropinocytotic (MP) events per area per time which we think is a more accurate reflection of the data.

Figure 2H. has a y-axis labeled “Ruffles fold change (log 2)” The legend says that this is the “quantification of the fold difference in macropinocytosis level...” This is quite a perplexing figure. Is it measuring ruffles or macropinosomes? What is meant by fold change (of what)? This is a very important plot and needs clarification and attention. Within this plot does each dot represent an embryo?

I would have expected something like # of macropinosomes/cell/time.

Figure 2H What was measured? How was it measured?

In this figure we presented the ratio between the number of MP events/area/time before and after the drug treatment for the same embryo (as a log₂ to ensure symmetry around zero). This was done because there is some embryo-to-embryo variability in the number of MP events and we wanted to make sure we were comparing the changes that occur within each experimental treatment. Each dot represents an embryo which was made clear in the 2h figure legend of the revised manuscript.

The dextran uptake assay under these inhibitors would be more compelling, but imaging of ruffles should suffice. Data showing examples of ruffle morphology under drug conditions should also be shown.

We provided a new movie (Movie S7) showing ruffle morphology and that each ruffle leads to dextran internalization under the drug treatments used in 2H. We did not observe any actin ruffle events that did not lead to dextran internalization.

Figure 3B – what are the halos colors above and below in the line plot? These seem far too small for variance. Individual traces should be shown.

This is greatly helped by plots C and D – which indicate that 3B does not display a meaningful variance.

For the original submission we used the difference between individual experiments as the variance, which we agree underrepresents the true variance. We have changed that for the resubmission in Fig3b to including the variance between all cells in all experiments which increases the variance substantially but does not alter the overall interpretation of the data. We also added new data (Fig. S1a) where we show individual traces from the smallest experiment, as the larger experiments, while consistent, became difficult to visualize due to the number of cells analyzed. Finally, it is worth noting that Fig3b uses normalized area while Fig3c uses raw area (μm^2).

Figure 3E –“ junction types labeled (N1-N4)” ...maybe... junction proximity (or edge-linkage) to the macropinosome? “Types” sounds vague.

We changed both the manuscript main text and Fig 3e to incorporate proximal/distal terminology regarding the N1/N2/N3/N4 junctions for more clarity.

Similarly, this statement (top of page 4) doesn't work for me “cell size reduction is primarily driven by internalization of a large portion of the apical membrane rather than internalization of the junctions per se.”

The apical area is shrinking. But the volume of the cell would increase owing to the internalization of extracellular fluid within the macropinosome. This seems to be more of a shape change – presumably creating an elongation (possibly infinitesimal) in the axes perpendicular to the plane of the junctions.

We have switched “cell size” to “apical size” throughout the manuscript to ensure accuracy.

Similar comments for the paragraph in the middle of page 4. There is no evidence that the volume of the cell decreases. In fact, the images in 1.H' appears to show f-actin near the basal surface extend downward after the macropinosome forms.

1.I' doesn't seem to show the basal membrane?

Could this be quantified? Granted this is the worst resolution direction and position of the confocal microscope.

We agree that apical size reduction could lead to cell elongation, unfortunately poor resolution at the basal level prevented us from accurately measuring either elongation or volume. Again we have switched the terminology to more accurately reflect the change to “apical size” only.

Similarly, in the 2D explant model, (Figure 2.D, E), part of the created tension would presumably be from the basal side of the cells pressed down against a rigid surface. A cell could only decrease its apical plane size by 1) getting taller (extrusion) or by 2)

expelling some its material. Reduction in cell size does not seem accurate. Reduction in apical surface does.

Agreed and changed.

Figure 4. The flip-flopping of color bars makes this figure almost intractable. Why flip the color bar from low-peach->high-blue for pressure and then invert it for tension? Wouldn't it follow that cells under lateral pressure would have lower mean edge tension? -> same colormap? Shear strain could also be shown a on a similar colormap.

We apologize for the confusing manner of our presentation. We have homogenized all colormaps for clarity. Also, the simulation shows cell tension rather edge tension (e.g. Fig 3e-f).

To make figure 4F-K more convincing, the data for MCC-adjacent high-tension cells should also be plotted. It appears that from the images and movies that MCC cells are enriching high- and low-tension cells. Over MCC-devoid regions.

MCC-adjacent cells are indeed also enriched in high tension cells, which we have added to the manuscript in Fig S2b. We would not necessarily expect a change in MP events in high tension cells relative to normal tension cells, but we agree that this is an important observation that should be reported.

The stepwise introduction of MCC cells in the simulation should also be described and interpreted in the results section for figure 4. The simulation appears to handle this change well, even though there is a noticeable shift in the compression and shear patterns, when they are introduced.

We have completely reworked this paragraph to better explain the MCC introduction and more importantly to expand on the interpretation.

The methods for compression in agarose gel are unclear – how long were the embryos embedded? Was any external force applied?

Embryos for this experiment were compressed solely by embedding them into 2% agarose, we clarified this point into the main text of the revised manuscript.

For the grease compression – can an estimate be given for how much the embryos were flattened? E.g. approximately how much were they compressed? This would be important for others to reproduce the work.

We agree that providing a quantitative estimate of embryo compression is important for reproducibility. To address this, we have now measured embryo height before and after compression to quantify the degree of flattening. Representative images, embryo height before and after as well as compression percentage are now in Fig S3c; S3d and S3e respectively.

“To evaluate overall tissue remodeling by the wave of macropinocytotic events, we measured absolute changes in cell size that considers both cell size reduction by macropinocytosis as well as cell stretching induced by events in the neighboring cells (Fig 5C-E, Movie S8).”

What is meant by absolute changes in cell size? This sounds like area and volume, but I don't think that is what was being measured.

We changed “cell size” to “apical size”. “Absolute” change refers to the change independent of direction (increase or decrease) as a measure of overall remodeling that can occur during the wave of MP events. This change can occur from apical size decrease in a cell having an MP event or apical size increase that would occur if numerous surrounding cells have an MP event. In the context of a wave of MP events some cells shrink their apical surface while some cells increase their apical surface (unless they are isolated in which case they always decrease their size Fig. S3g).

Reviewer #3:

In Fig. 4A and B, Roscovitine is used to block cell proliferation/crowding, yet more MCCs are observed. It will be valuable to include cell density measurement together with MCCs here to show that before and after treatment, the number of MCCs do not change, but the loss of MCCs in control embryos is prevented under the treatment condition.

The effect of Roscovitine on cell density is indeed an important control that we did not originally include. We have now added goblet cell density upon Roscovitine in Fig 2c.

In Fig. 4M, the pattern of MCCs in LY294002-treated embryos seems to be different from that of DMSO- (change the label “Control”) or Amiloride-treated embryos. Is this true? Please provide some explanation about this.

This specific pattern does not appear consistently with LY294002 treatment overall and was a poor choice of “representative” embryo. We have switched out the image in Fig 4m with a more consistent representative image that still shows the same phenotype on overall MCC number. We have also relabeled the control as DMSO.